# Closed Boundary Learning for Classification Tasks with the Universum Class

**Hanzhang Zhou**[1,3]**, Zijian Feng**[1,3] **Kezhi Mao**[2,3*]

[1]Institute of Catastrophe Risk Management, Interdisciplinary Graduate Programme,
Nanyang Technological University, Singapore
[2]School of Electrical and Electronic Engineering, Nanyang Technological University, Singapore
[3]Future Resilient Systems Programme, Singapore-ETH Centre, CREATE campus, Singapore
{hanzhang001, feng0119}@e.ntu.edu.sg
ekzmao@ntu.edu.sg

## Abstract

The Universum class, often known as the *other* class or the *miscellaneous* class, is defined as a collection of samples that do not belong to any class of interest. It is a typical class that exists in many classification-based tasks in NLP, such as relation extraction, named entity recognition, sentiment analysis, etc. The Universum class exhibits very different properties, namely heterogeneity and lack of representativeness in training data; however, existing methods often treat the Universum class equally with the classes of interest, leading to problems such as overfitting, misclassification, and diminished model robustness. In this work, we propose a closed boundary learning method that applies closed decision boundaries to classes of interest and designates the area outside all closed boundaries in the feature space as the space of the Universum class. Specifically, we formulate closed boundaries as arbitrary shapes, propose the inter-class rule-based probability estimation for the Universum class to cater to its unique properties, and propose a boundary learning loss to adjust decision boundaries based on the balance of misclassified samples inside and outside the boundary. In adherence to the natural properties of the Universum class, our method enhances both accuracy and robustness of classification models, demonstrated by improvements on six state-of-the-art works across three different tasks. Our code is available at https://github.com/hzzhou01/Closed-Boundary-Learning

## 1 Introduction

In classification-based tasks, quite often we encounter a class named as *other* class, *miscellaneous* class, *neutral* class or *outside (O)* class. Such a class is a collection of samples that do not belong to any class of interest, such as samples of *no relation* class in relation extraction task. We adopt the terminology in (Weston et al., 2006) to

---

*Corresponding author.

designate all such classes as the *Universum class* (U). Universum class exits in various classification-based problems in NLP, such as relation extraction (RE) (Zhang et al., 2017), named entity recognition (NER) (Tjong Kim Sang and De Meulder, 2003), sentiment analysis (SA) (Tjong Kim Sang and De Meulder, 2003), and natural language inference (NLI) (Bowman et al., 2015). To distinguish the Universum class and the rest of the classes, we call the classes of interest as *target classes* (T). The set of all classes (A) in the data can be expressed as $A = U \cup T$

- *Universum class*: A collection of samples that do not belong to any class of interest.

- *Target class*: A class of interest in the task, i.e., one of the classes other than the Universum class.

The sample compositions of the Universum class and target classes are usually very different. Figure 1(a) provides some samples of a target class (*entity-destination*) and the Universum class (*other*) in relation extraction. Intuitively, we can observe that the *entity-destination* samples adhere to an **intra-class pattern**: an entity goes somewhere. However, the three examples of the *other* relation type are vastly dissimilar and do not exhibit any intra-class pattern. In fact, the Universum samples are labeled according to an **inter-class pattern**: they do not belong to any of the predefined target classes.

We further highlight the differences between the Universum class and target classes in two properties.

(1) **Heterogeneity**: The Universum class is composed of heterogeneous samples, which may form multiple clusters in the feature space of the test set, as illustrated by the green samples in Figure 1(c). This is because the Universum class, as the class name "other" implies, contains all potential implicit classes that are not explicitly defined in the

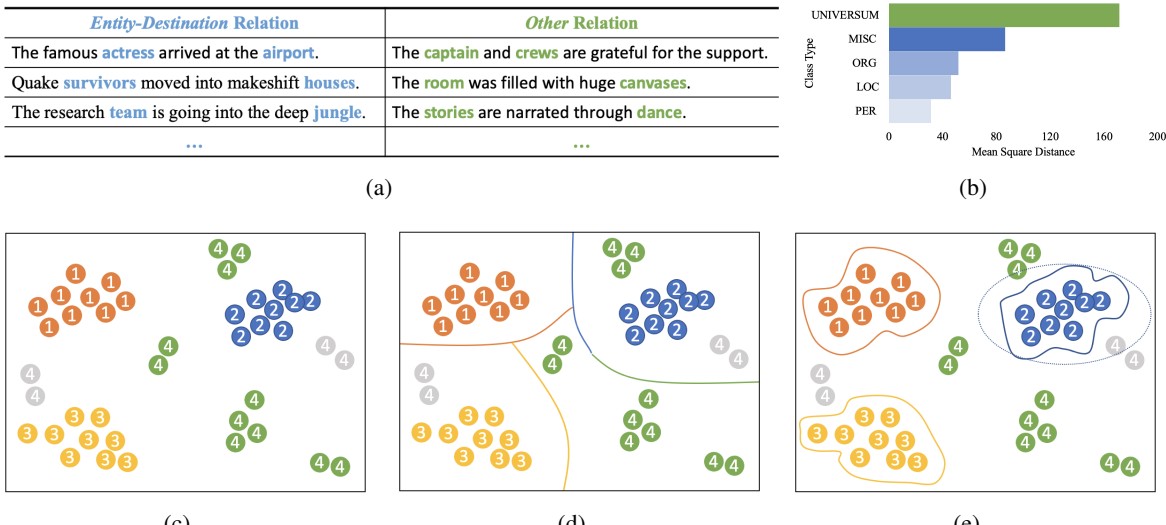

Figure 1: Illustration of distinction between the Universum class. (a) Samples selected from the SemEval 2010 Task 8 dataset on relation extraction. (b) Compactness comparison between the test data of the Universum class (green) and target classes (blue). (c) The distribution of target classes (class 1, 2, 3) and the Universum class (class 4). In particular, the gray samples represent Universum samples in the test set that are not represented by the training data. (d) The open decision boundaries obtained by traditional classifiers. (e) The arbitrary closed boundaries obtained by our proposed method.

task. For example, in samples of the *other* class in Figure 1(a), implicit classes may include the entity-parallel relationship, the entity-fill relationship, and the entity-narrative relationship.

Although such heterogeneous samples are easily mapped into a compact cluster for the training set, it is problematic for the test set. This is because the inherent predictive rule of the Universum class follows an unique *inter-class* pattern: a sample is labeled as Universum if it does not belong to any target classes. This sharply contrasts with the conventional *intra-class* patterns seen in target classes. Considering human annotation practices, an entity is labeled as *Location* when it aligns with established patterns of *Location* entities. In contrast, a sample is labeled as *Others* not due to intra-class patterns specific to the *Others* class, but because it fails to conform to the patterns of *Location*, *Person*, or *Organization*. Consequently, when current classification models treat the Universum class and target classes in the same manner, they tend to overfit the noise in the Universum class by memorizing various peculiarities of the heterogeneous samples rather than recognizing the general predictive rule. Given the variations in data distributions between the test and training sets, only memorizing various peculiarities can easily lead to *overfitting*, causing a decline in accuracy. Furthermore, this inabil-

ity to discern the genuine predictive rule for the Universum class can also compromise the model's robustness.

(2) **Lack of Representativeness in Training Data**: The Universum class is the complementary set of predefined target classes in the task. Therefore, it contains all possible implicit classes, i.e., classes not explicitly defined in the task but may appear in the real world. In this case, Universum samples in the training data are unable to sufficiently represent all possible patterns of the genuine distribution of the Universum class. As depicted in Figure 1(c), gray samples represent Universum samples in the test set that are not represented by the training data. Classifiers with open boundaries are prone to misclassifying unseen samples in the test set that is not represented by the training data.

Additionally, we provide a quantitative comparison of the average compactness between the Universum class and the target classes within the test data for the NER task (Fu et al., 2021), as depicted in Figure 1(b). Notably, even though the Universum class is the class with the most samples, it exhibits significantly poorer compactness in its learned representations. This empirical observation aligns with our earlier theoretical analysis. Both the inability to discern the genuine *inter-class* predictive rule of the Universum class and the lack of

representativeness in the training data contribute to this compromised compactness for the Universum class. Experiment details are in Appendix A.

Despite the substantial difference between the target classes and the Universum class, this issue has long been neglected by the research community. The majority of works (Zhu and Li, 2022; Ye et al., 2022; Wan et al., 2022; Fu et al., 2021; Li et al., 2021b) treat the Universum class and target classes equally. Typically, a linear layer and a softmax function are applied at the end of the model to generate open decision boundaries, which we believe are inappropriate for tasks containing the Universum class.

How can we account for the distinct properties of the Universum class and target classes to derive better representations and classifiers? We think the key lies in **conforming to the inherent properties of the Universum class**. In this work, we propose a closed boundary learning method for classification-based tasks with the Universum class. Traditional methods often employ open boundary classifiers and constrain the representations of Universum samples to be distributed into a compact cluster. However, the open decision boundaries can easily misclassify Universum samples, as illustrated in Figure 1(d). In addition, the restriction on compact space violates the inherent inter-class pattern of the Universum class. Therefore, we propose to use closed boundary classifiers as shown in Figure 1(e). We constrain the space of target classes to be closed spaces and designate the area outside all closed boundaries in the feature space as the space of the Universum class. The treatment perfectly fits the nature of the Universum class: a sample is marked as the Universum if it does not belong to any target class during labeling.

The main contributions of this work are summarized as follows: (1) We address an understudied problem in this paper. The Universum class widely exists in many NLP tasks and general machine learning tasks, but hasn't received significant attention in these contexts. (2) Methodologically, we generate closed boundaries with arbitrary shape, propose the inter-class rule-based probability estimation for the Universum class to cater to the inherent properties of the Universum class, and propose a boundary learning loss to learn the decision boundary based on the balance of misclassified samples inside and outside the boundary. (3) In adherence to the natural properties of the Univer-

sum class, our method improves both accuracy and robustness of classification models, which is validated on six state-of-the-art (SOTA) works across three different tasks.

## 2 Related Works

### 2.1 Classification Tasks with the Universum Class

The Universum class widely exists in classification based tasks in NLP, such as relation extraction (RE) (Zhang et al., 2017), named entity recognition (NER) (Tjong Kim Sang and De Meulder, 2003), and aspect category sentiment analysis (ACSA) (Jiang et al., 2019), as summarized in Table 1. It should be noted that the span-based methods (Zhu and Li, 2022; Li et al., 2021a) enumerate all possible spans for classification, which introduces an extra *other* class. Despite the heterogeneity and lack of representativeness of the Universum class, current works (Zhu and Li, 2022; Wan et al., 2022; Fu et al., 2021; Tian et al., 2021; Chen et al., 2021; Li et al., 2021b, 2020; Yu et al., 2020) solve the classification problems containing the Universum class as normal multi-class classification problems and treat the Universum class and target classes equally.

### 2.2 Closed Boundary Learning Methods

Closed boundaries are often adopted in research fields of out-of-distribution (OOD) detection (Gomes et al., 2022; Ren et al., 2021; Chen et al., 2020), open set recognition (Zhang et al., 2021; Liu et al., 2020), anomaly detection (Zong et al., 2018), and outlier detection (Sharan et al., 2018; Sugiyama and Borgwardt, 2013). We borrow the term "generalized OOD detection" from (Yang et al., 2021b) to encapsulate these problems. Furthermore, we summarize the differences between the classification with the Universum problem and the generalized OOD detection, focusing on both problem setting and methodology. The detailed analysis are provided in Appendix B.

### 2.3 Universum Learning Methods

Although we adopt the terminology of Universum (Weston et al., 2006), the problem setting of our work is entirely different from that of previous studies on Universum learning (Weston et al., 2006; Chapelle et al., 2007; Qi et al., 2012; Zhang and Le-Cun, 2017). The idea of Universum learning studies is to exploit external, unlabeled Universum data

| Task | Dataset | Label Name | Proportion |
|---|---|---|---|
| RE | SemEval 2010 Task 8 (Hendrickx et al., 2019) | Other | 17.4% |
| RE | TARCED (Zhang et al., 2017) | No relation | 79.5% |
| NER | CoNLL-2003 (Tjong Kim Sang and De Meulder, 2003) | Miscellaneous | 14.6% |
| NER (span based method) | CoNLL-2003 (Tjong Kim Sang and De Meulder, 2003) | Other | >90% |
| ACSA | MAMS (Jiang et al., 2019) | Neutral | 43.4% |

Table 1: The tasks and datasets that the Universum class exists.

to improve the accuracy of supervised tasks. However, in our problem, the Universum class is one of the internal, labeled classes of multi-class classification problems and we propose closed boundary learning to conform to its unique properties during learning. Furthermore, methodologically, the Universum learning method either employs open-boundary classifiers (Zhang and LeCun, 2017) or is incapable of distinguishing the Universum samples from labeled samples (Weston et al., 2006; Chapelle et al., 2007; Qi et al., 2012), neither of which are appropriate to the problem we presented.

## 3 Method

**Problem Definition**: The goal of our proposed method is to learn closed decision boundaries for target classes and meanwhile, jointly classify the Universum samples and target samples. We first give a detailed description of how to recognize the Universum class in Appendix C. In order to make our proposed method compatible with most existing classification methods, the starting point of our method is the representations of the final layer of classification models, which is a linear layer that maps data from high-dimensional feature space to a lower-dimension space. We denote the sample representations of the final linear layer as $\mathrm{H} = \{\mathbf{h}_0, \mathbf{h}_1, \dots, \mathbf{h}_{N-1}\} \in \mathbb{R}^{N \times l}$, where $N$ is the number of samples, and $l$ is the output dimension of the linear layer.

### 3.1 Pretraining

Our method estimates the probability distribution of target classes based on their sample distributions. In order to avoid estimation based on randomly initialized weight and speed up the learning process, we employ N-pair loss (Sohn, 2016) for pretraining, making sample representations be of small intra-class distance and large inter-class distance. Notably, in accordance with the nature of the Universum class, we make a change that does not require the model to reduce the intra-class distance

of Universum samples during the pretraining.

### 3.2 Generating Closed Boundary of Arbitrary Shape for Target Classes

Existing closed boundary classification methods mainly use spherical shape boundaries (Zhang et al., 2021; Liu et al., 2020); however, we argue that the spherical shape may not be the optimal solution because data samples are unlikely to perfectly fit into a sphere, and a spherical shape boundary may produce misclassifications. We adopt the Gaussian mixture model (GMM) and the threshold value to generate boundaries with arbitrary shapes.

#### 3.2.1 Gaussian Mixture Model

We apply GMM with $m$ components to estimate the class conditional probability distribution for each target class $C_i$, and further derive the joint probability estimation for each class.

$$p(\mathbf{h}_k \mid C_i) = \sum_{i=1}^{m} \pi_i \mathcal{N}\left(\mathbf{h}_k; \boldsymbol{\mu}_i, \boldsymbol{\Sigma}_i\right) \quad (1)$$

$$p(\mathbf{h}_k, C_i) = p(\mathbf{h}_k \mid C_i)p(C_i) \quad (2)$$

where $\mathbf{h}_k$ denotes the input feature vector of the $k$th sample, $\boldsymbol{\mu}_i$ and $\boldsymbol{\Sigma}_i$ are the estimated mean vector and covariance matrix of the $i$th Gaussian components, respectively. $\pi_{ij}$ is the non-negative mixture weight under the constraint that $\sum_{j=1}^{m} \pi_{ij} = 1$. $\boldsymbol{\mu}_i, \boldsymbol{\Sigma}_i$, and $\pi_{ij}$ are all learnable parameters in the model.

According to Bayes Theorem, the posterior probability $p(C_i \mid \mathbf{h}_k) = \frac{p(\mathbf{h}_k|C_i)p(C_i)}{p(\mathbf{h}_k)}$. Since we are interested in $\mathrm{argmax}_{C_i} \frac{p(\mathbf{h}_k|C_i)p(C_i)}{p(\mathbf{h}_k)}$, the decision can be made based on joint probability $p(\mathbf{h}_k, C_i)$.

#### 3.2.2 Arbitrary Shape Boundary

**Geometrical View**: Inspired by the DENCLUE algorithm in generating arbitrary shape clusters (Hinneburg and Keim, 1998), we introduce a threshold value $\xi_i$ for each target class. A closed boundary of arbitrary shape is formulated by points satisfying $p(\mathbf{h}, C_i) = \xi_i$. Figure 2 is an illustration of

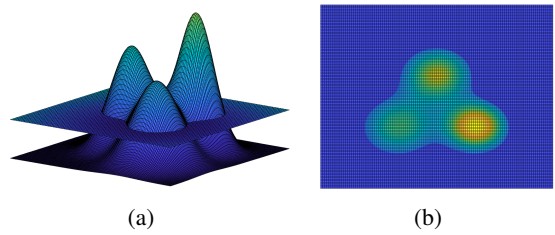

(a)         (b)

Figure 2: Illustration of generating arbitrary shape boundaries.

formulating an arbitrary shape boundary in a two-dimensional space. A sample is assigned to class $C_i$ if it is located inside the closed boundary. If the number of components of the GMM is set to one, then the shape of the boundary becomes spherical, with its center and covariance matrix specified by $\mu_0$ and $\Sigma_0$, respectively. In this sense, the commonly used spherical shape boundary (Zhang et al., 2021; Liu et al., 2020) is a special case of our method.

Notably, the threshold values $\Xi = \xi_1, \xi_2, \cdots, \xi_{n-1}$ are a learnable parameters, which eliminates the laborious process of hyperparameter tunning. Specifically, they are learned based on the balance of misclassified samples inside and outside the boundary through our proposed boundary learning loss, which is introduced later.

**Probabilistic View**: The above geometrical process can be described as:

$$\begin{cases} \mathbf{h}_k \in C_i \text{ if } p(\mathbf{h}_k, C_i) > \xi_i \\ \mathbf{h}_k \notin C_i \text{ if } p(\mathbf{h}_k, C_i) \leq \xi_i \end{cases} \quad (3)$$

### 3.3 Inter-Class Rule-Based Probability Estimation for the Universum Class

The main obstacle to properly addressing the issue of the Universum class is to estimate the probability of the Universum class based on its inherent *inter-class* property rather than *intra-class* sample distributions. We propose an inter-class rule-based probability estimation method to address this issue.

#### 3.3.1 Motivation and the Estimation

We classify samples of Universum class and target classes based on the following rules we defined:

- *Rule 1*: A sample is assigned to the Universum class if it is not located inside any of the closed boundaries of target classes.

- *Rule 2*: A sample is assigned to the target class with the highest $p(\mathbf{h_k}, C_i)$ if it is located inside at least one closed boundary.

An intuitive way to incorporate the above rules is a two-step method that consists of Universum class detection and target class classification. However, such a pipeline method has the issue of error propagation. In addition, general probability estimation methods exploit intra-class sample distributions, which fail to overcome the natural inter-class property of the Universum class and do not conform to Rule 1. Therefore, a strategy need to be devised to convert Rule 1 and 2 into a probability expression, while simultaneously facilitates the learning of the neural network.

For compliance with Rule 1, the estimated probability of the Universum class must satisfy the following two conditions: for Universum class samples: $\forall i : p(\mathbf{h}_k, U) > p(\mathbf{h}_k, C_i)$ and for target class samples: $\exists i : p(\mathbf{h}_k, U) < p(\mathbf{h}_k, C_i)$. We can leverage the relationship between $\xi_i$ and $p(\mathbf{h}_k, C_i)$ defined in Equation 3 to construct the estimation of $p(\mathbf{h}_k, U)$ that satisfies the above two conditions. In addition, to enhance the learning of neural networks, the gradient obtained from an Universum sample should move this sample away from its closest target class boundary. Therefore, we also involve $\max(p(\mathbf{h}_k, C_i))$, the probability of the closest target class of a Universum sample, to guide the Universum sample move away from target class boundaries. We propose to estimate the probability distribution of the Universum class as follows:

$$p(\mathbf{h}_k, U) = \lambda \frac{\xi_u^2}{p(\mathbf{h}_k, C_u)} + (1 - \lambda) \frac{\xi_v^2}{p(\mathbf{h}_k, C_v)} \quad (4)$$

$$\text{where} \quad \lambda = \begin{cases} 1, \ p(\mathbf{h}_k, C_u) > \xi_u \\ 0, \ p(\mathbf{h}_k, C_u) \leq \xi_u \end{cases} \quad (5)$$

$$\begin{cases} u = \text{argmax}_i \frac{p(\mathbf{h}_k, C_i)}{\xi_i}, \\ v = \text{argmax}_i p(\mathbf{h}_k, C_i) \end{cases} \quad (6)$$

$\xi_i$ is the threshold value of target class i, and $u, v \in \{1, 2, \ldots, n-1\}$.

#### 3.3.2 Analysis of the Proposed Estimation

For estimated probability of the Universum class in Equation 4, two cases are possible.

**Case 1**: $p(\mathbf{h}_k, C_u) > \xi_u$, i.e., sample $\mathbf{h}_k$ is located inside at least one closed boundary.

In this case, we have

$$p(\mathbf{h}_k, U) = \xi_u \frac{\xi_u}{p(\mathbf{h}_k, C_u)} < \xi_u < p(\mathbf{h}_k, C_u)$$

Since $p(\mathbf{h}_k, U) < p(\mathbf{h}_k, C_u)$, the model will select the target class i with the highest $p(\mathbf{h}_k, C_i)$, which

fits perfectly with Rule 2.

**Case 2**: $p(\mathbf{h}_k, C_u) \leq \xi_u$, i.e., the sample $\mathbf{h}_k$ distribute outside every closed boundary.

Combining the condition of case 2 and Equation 6, we have

$$\forall i \in \{1, 2, \ldots, n-1\} : \frac{p(\mathbf{h}_k, C_i)}{\xi_i} \leq \frac{p(\mathbf{h}_k, C_u)}{\xi_u} \leq 1$$

$$\text{i.e.,} \forall i \in \{1, 2, \ldots, n-1\} : p(\mathbf{h}_k, C_i) \leq \xi_i \quad (7)$$

Combining Equation 6 and Equation 7, we can derive that $\forall i \in \{1, 2, \ldots, n-1\}$:

$$p(\mathbf{h}_k, U) = \xi_v \frac{\xi_v}{p(\mathbf{h}_k, C_v)} \geq \xi_v \geq p(\mathbf{h}_k, C_v) \geq p(\mathbf{h}_k, C_i)$$

In case 2, from Equation 7 and Equation 3, we can learn that sample $\mathbf{h}_k$ is located outside all closed boundaries of target classes. In this case, the probability of Universum class $p(\mathbf{h}_k, U)$ obtains the largest value. Therefore, Rule 1 is perfectly expressed by the proposed probability estimation of the Universum class.

### 3.4 Boundary Learning Loss

To facilitate the learning of the closed decision boundaries, we propose a boundary learning loss below. Our intuition is that the decision boundary should be adjusted to the balance of misclassified samples inside and outside the boundary. For example, if samples of class $j$ distribute inside the boundary of class $i$, then the boundary should contract to exclude such samples and vice versa.

$$L_{bl} = \frac{1}{M} \sum_{i=1}^{n-1} \left( \sum_{k \in \mathbb{O}} w_k \log \frac{\xi_i}{p(\mathbf{h}_k, C_i)} \right.$$
$$\left. + \sum_{l \in \mathbb{I}} w_l \log \frac{p(\mathbf{h}_l, C_i)}{\xi_i} \right)$$

$M$ is the total number of misclassified samples for all boundaries, $n$ is the number of classes, $\mathbb{O}$ and $\mathbb{I}$ denote the set of training samples misclassified outside and inside the decision boundary $i$, respectively. The weights in the loss function are $w_k = \frac{p(\mathbf{h}_k, C_i)}{p(\mathbf{h}_k, C_i) + \xi_i}$, $w_l = \frac{\xi_i}{p(\mathbf{h}_l, C_i) + \xi_i}$, and they are detached and cut off the gradient. Weights $w_k$ and $w_l$ have smaller values for samples located far from the boundary, enabling the boundary to be adjusted primarily on the basis of easily and semi-hard negatives instead of hard negatives.

During training, we sum the cross-entropy loss and boundary learning loss for optimization. In addition to balancing inside and outside misclassified samples, the boundary learning loss forces misclassified samples to be distributed in the proper region, which works well with cross-entropy loss.

### 3.5 Framework Overview

To provide better clarity of our method, we provide a succinct breakdown of the closed boundary learning framework's workings:

- **Initialization Post-Pretraining**: After pretraining, we employ the GMM for each target class. The Expectation-Maximization (EM) algorithm is employed to set the initial values for the GMM parameters $\boldsymbol{\mu}_i$, $\boldsymbol{\Sigma}_i$, and $\pi_{ij}$. As we transition to the training phase, the parameters of GMM are treated as learnable parameters. Contrary to traditional methods using the EM algorithm for continuous updates, these parameters are dynamically updated by the neural network throughout the training process.

- **Probability Estimation**: Probability distributions of target classes are estimated using GMM, as articulated in Equation 2. The probability distribution of the Universum class is computed through our Inter-Class Rule-Based Probability Estimation method, which is represented in Equation 4.

- **Training Optimization**: During the training process, we use a combined loss function, summing the cross-entropy loss with the boundary learning loss for optimization.

## 4 Experiments

### 4.1 Experimental Methodology

We demonstrate the efficacy of our method on six different SOTA models on three datasets of different NLP tasks, including SemEval 2010 Task 8 (Hendrickx et al., 2019), MAMS (Jiang et al., 2019), and CoNLL-2003 (Tjong Kim Sang and De Meulder, 2003). The proportion of Universum samples in the SemEval 2010 Task 8, MAMS, and CoNLL-2003 datasets are 17.4% (highest in 19 classes), 90%, and 43.4%. respectively. It is noteworthy that the ratio of Universum class in the NER task is not calculated from the *miscellaneous* samples in the dataset but from the *other* samples which are introduced by the span-based method (Zhu and Li, 2022; Fu et al., 2021).

We evaluate the effectiveness of our proposed

| Task | Method | F1/accuracy | p-value |
|------|--------|-------------|---------|
| NER | SpanNER (Fu et al., 2021) | 92.09±0.16 | |
| | SpanNER (Fu et al., 2021) + ADB | 77.22 ± 0.49 | < 0.001 |
| | SpanNER (Fu et al., 2021) + OECC | 91.22 ± 0.12 | |
| | SpanNER (Fu et al., 2021) + COOLU | **93.50**±0.13 | |
| | BS (Zhu and Li, 2022) | 92.53±0.02 | |
| | BS (Zhu and Li, 2022) + ADB | 75.52 ± 0.55 | < 0.01 |
| | BS (Zhu and Li, 2022) + OECC | 91.88 ± 0.15 | |
| | BS (Zhu and Li, 2022) + COOLU | **93.17**±0.13 | |
| RE | A-GCN (Tian et al., 2021) | 88.67±0.18 | |
| | A-GCN (Tian et al., 2021) + ADB | 85.99 ± 0.23 | < 0.01 |
| | A-GCN (Tian et al., 2021) + OECC | 88.00 ± 0.09 | |
| | A-GCN (Tian et al., 2021) + COOLU | **89.33**±0.20 | |
| | TaMM (Chen et al., 2021) | 88.76±0.23 | |
| | TaMM (Chen et al., 2021) + ADB | 85.08 ± 0.26 | < 0.01 |
| | TaMM (Chen et al., 2021) + OECC | 88.17 ± 0.18 | |
| | TaMM (Chen et al., 2021) + COOLU | **89.47**±0.21 | |
| ACSA | AC-MIMLLN (Li et al., 2020) | 76.13±0.29% | |
| | AC-MIMLLN (Li et al., 2020) + ADB | 71.78 ± 0.85% | < 0.01 |
| | AC-MIMLLN (Li et al., 2020) + OECC | 74.02 ± 0.68% | |
| | AC-MIMLLNN (Li et al., 2020) + COOLU | **77.35**±0.42% | |
| | SCAPT (Li et al., 2021b) | 84.13±0.19% | |
| | SCAPT (Li et al., 2021b) + ADB | 79.67 ± 0.44% | < 0.01 |
| | SCAPT (Li et al., 2021b) + OECC | 83.36 ± 0.27% | |
| | SCAPT (Li et al., 2021b) + COOLU | **85.06**±0.23% | |

Table 2: The overall performance of applying closed boundary learning on baseline models.

**Cl**O**sed b**O**undary **L**earning for classiciation with the **U**niversum class (**COOLU**) on 6 SOTA works, including SpanNER (Fu et al., 2021), BS (Zhu and Li, 2022), A-GCN (Tian et al., 2021), TaMM (Chen et al., 2021), AC-MIMLLN (Li et al., 2020), and SCAPT (Li et al., 2021b). The implementation details are in the Appendix D.

### 4.2 Overall Experimental Results

Our first research question (RQ) is *can COOLU achieve a "free" accuracy gain on tasks with the Universum class?* (**RQ1**) Table 2 shows the overall results for all 6 models. The reported results are the average of three runs. Models with our proposed closed boundary learning outperform the original models with open classifiers on NLP tasks containing the Universum class. The overall accuracy or F1 score is improved on all six models we evaluated, with the largest improvement from 92.09 to 93.50 in F1 score. We also notice that the improvement on the RE task is not as significant as on the NER and the ACSA tasks (0.66 against 1.41 and 1.22). This may be due to the fact that Universum samples only account for 17.4% of the SemEval 2010 Task 8, which is considerably less than the other datasets. In addition, statistical tests between the accuracy/F1 score of the baseline models and our method indicate that the improvement brought

about by our COOLU method is statistically significant. The above experimental results answer **RQ1** in positive.

### 4.3 A Closer Look at the MicroF1, Precision and Recall

Another question is *does COOLU enhance classification accuracy for all classes or just the Universum class?* (**RQ2**) We show the micro F1 score of each class in applying closed boundary learning on SpanNER (Fu et al., 2021) in Table 3. The micro F1 score for the Universum (*other*) class, introduced by the span-based method, is excluded from the overall F1 score calculation as per the task requirements. The micro F1 score is improved in all classes, with the absolute improvement of 0.01, 2.43, 0.29, 1.31, and 4.02, respectively. The F1 improvement of the Universum class is very small compared to target classes because its sample size is more than 100 times larger than other classes, making the denominator very large when calculating. The results answer **RQ2** positively: the improvement of overall performance is not only attributed to the improvement of the Universum class, but also to the improvement of all classes as a whole.

The third research question is *how does COOLU improve classification model's performance?* (**RQ3**) We find that our proposed closed

| Method | ORG | | | PER | | | LOC | | | MISC | | | Other | | |
|---|---|---|---|---|---|---|---|---|---|---|---|---|---|---|---|
| | P | R | F1 | P | R | F1 | P | R | F1 | P | R | F1 | P | R | F1 |
| SpanNER | 89.92 | 89.81 | 89.87 | 97.74 | 96.47 | 97.11 | 93.05 | 93.88 | 93.46 | 78.99 | 82.83 | 80.87 | 99.87 | 99.83 | 99.85 |
| SpanNER + COOLU | **94.29** | 90.30 | **92.30** | **98.54** | 96.29 | **97.40** | **96.17** | 93.41 | **94.77** | **88.02** | 81.97 | **84.89** | 99.84 | **99.89** | 99.86 |

Table 3: The micro F1 score of SpanNER (Fu et al., 2021) with and without closed boundary learning.

boundary learning significantly improves the precision score of all target classes, with the largest absolute gain being 9.03 in Table 3. By analyzing the change in precision and recall, we can derive the following findings: Firstly, the misclassification of Universum samples as target samples results in low precision scores for target classes and low recall score for the Universum class in the baseline method, which proves our claim that the Universum class is easily misclassified if its unique properties are neglected. In addition, our proposed closed boundary learning method can effectively prevent the misclassification of the Universum class into target classes, which significantly improve the precision score of target classes at the expense of a very slight reduction in recall and similarly improve the recall score of the Universum class at the expense of a slight reduction in precision. The second finding answers **RQ3**.

## 4.4 Model Robustness Evaluation

We are more interest in the research question that *In adherence to the natural properties of the Universum class, does COOLU provide a more reasonable way of learning by improving both model's accuracy and robustness?* (**RQ4**) We attempt to demonstrate RQ4 by theoretical analysis and experimental evaluation of the robustness of the model. Theoretically, since the natural predictive rule of the Universum class is an *inter-class* pattern that does not belong to any target classes, traditional models are more likely to fit the noise in the Universum class by memorizing various peculiarities of *intra-class* heterogeneous samples rather than finding the general predictive rule. However, our method can identify the inter-class predictive rule of the Universum class and hence classify out-of-distribution Universum samples more accurately. In addition, the closed decision boundaries we learned are analogous to model's knowledge boundary of each target class: the space inside the boundary represents what the model knows about a certain class, i.e., the recognized patterns, whereas the space outside the boundary represents what the model doesn't know about this class from training

data. Such knowledge boundary can avoid the misclassification of unseen non-target class samples as target class. The above two mechanism would contribute to better robustness of the model.

We evaluate the robustness of the model based on TextFlint (Wang et al., 2021), a robustness evaluation toolkit for NLP tasks. Specifically, TextFlint generate perturbations of the test data, and the robustness of the model is evaluated using the transformed test dataset. The terms such as "CrossCategory", "OOV", and "SpellingError" in Table 4 are different ways of transforming the test data. Detail information of these transformation methods are illustrated in Appendix D.3.

It can be learned from the Table 4 that the robustness of SpanNER improved significantly by applying our proposed COOLU method, with the improvement of absolute F1 score of 4.44, 8.01, 5.03, and 2.55, respectively. In addition, although the improvement of the F1 score in A-GCN is less than 1, the improvement in robustness of the model is considerably large. The absolute F1 score on robustness evaluation datasets are improved at 2.33, 1.89, 1.35, and 1.63, respectively. Considering lots of studies theoretically identify a trade-off between robustness and accuracy (Tsipras et al., 2019; Zhang et al., 2019; Raghunathan et al., 2020), the improvement of both model's accuracy and robustness provides positive evidence for **RQ4**: COOLU can provide a more reasonable way of learning representations and classifiers.

## 4.5 Ablations

We evaluated two recent OOD detection methods, ADB (Zhang et al., 2021) and OECC (Papadopoulos et al., 2021), to demonstrate the inappropriateness of OOD detection approaches for our proposed problem. The experimental results reveal that integrating ADB and OECC with classification models considerably weakens the performance of the original models, as illustrated in Table 2. Given that both ADB and OECC were primarily designed for OOD detection and not the unique problem presented in this work, their unsuitability is anticipated. Specifically, due to the inher-

|  | CrossCategory | OOV | SpellingError | AppendIrr |
|---|---|---|---|---|
| SpanNER | 77.06 | 75.14 | 76.09 | 87.34 |
| SpanNER + COOLU | **81.39** | **83.15** | **81.12** | **89.89** |
|  | InsertClause | SwapEnt | SpellingError | AppendIrr |
| A-GCN | 77.84 | 86.8 | 77.36 | 86.59 |
| A-GCN + COOLU | **80.17** | **88.69** | **78.71** | **88.22** |

Table 4: Comparison of model's robustness with and without closed boundary learning.

| Method | F1/accuracy |
|---|---|
| SpanNER | 92.09 |
| SpanNER + N-pair pretraining | 92.05 |
| SpanNER + COOLU (N-pair pretraining) | **93.50** |
| SCAPT | 84.13% |
| SCAPT + N-pair pretraining | 84.16% |
| SCAPT + COOLU (N-pair pretraining) | **85.06**% |

Table 5: The effect of pretraining on SpanNER (Fu et al., 2021) and SCAPT (Li et al., 2021b).

ent difference in the problem settings, the ADB method cannot utilize the label information of the Universum class, leading to a significant decline in accuracy, especially when the Universum samples constitute a large portion in the span-based method. In addition, while OECC's problem setting, outlier exposure, aligns more closely with ours compared to other general OOD detection methods, it still shows inappropriateness, diminishing the accuracy of original classification models. This diminished accuracy can be attributed to: (1) error propagation in its two-step approach, and (2) the instability that arises from manually set thresholds.

We also conduct an ablation study on N-pair loss pretraining. In our method, N-pair loss is adopted for pretraining to learn initial representations and to speed up the training process. To demonstrate the effectiveness of our closed boundary learning method and to rule out the possibility that the improvement of our model is due to the N-pair loss pretraining process, we add an additional pretraining step to baseline models of SpanNER (Fu et al., 2021) and SCAPT (Li et al., 2021b). Table 5 indicates that the pretraining process alone cannot improve the accuracy of the original baseline models. The improvement brought about by our proposed closed boundary learning is not a result of the pretraining step but the result of the entire system.

In addition, an ablation study on final layer dimension is illustrated in Appendix E.

## 5 Conclusion

In this work, we highlight an understudied problem in classification-based tasks that the Universum class is treated equally with target classes despite their significant differences. As a solution, we propose a closed boundary learning method COOLU, which conforms the natural properties of the Universum samples. Specifically, we generate closed boundaries with arbitrary shapes, develop an inter-class rule-based strategy to estimate the probability of the Universum class, and propose a boundary learning loss to adjust decision boundaries. COOLU offers easy integration with most classification model, given that it operates on representations of the final layer of classification models. Our method not only boosts the accuracy of SOTA models but also significantly enhances their robustness.

## 6 Limitations

As a limitation, our method is not suitable for zero-shot or few-shot settings because the accuracy of GMM estimation is positively related to the number of samples used (Psutka and Psutka, 2019). Similarly, due to the inherent low-dimension constraints of the GMM, an extensive increase in the number of classes could pose challenges to the efficacy of our framework. Nevertheless, given that the last layer dimension, i.e. class number, in our method is usually small and the initialized GMM parameters will be fine-tuned by the neural network, most classification tasks are not limited by these constraints. Since the Universum class widely exists in NLP tasks and many general ML tasks, our method is applicable to most of these tasks.

### Acknowledgements

The authors would like to thank Edmond Lo, Lihui Chen, Xiyu Zhang, Zixiao Zhu, and the anonymous reviewers for their constructive comments and suggestions. The research was conducted at the Future Resilient Systems at the Singapore-ETH Cen-

tre, which was established collaboratively between ETH Zurich and the National Research Foundation Singapore. This research is supported by the National Research Foundation Singapore (NRF) under its Campus for Research Excellence and Technological Enterprise (CREATE) programme.

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

## A    Compactness of the Universum Class of the Test Set

We evaluate the compactness of the Universum class and target classes on the test data and depict the result in Figure 3. The representation of test samples after learning with open boundary classifiers by SpanNER (Fu et al., 2021) is used for evaluation. We evaluate the compactness based on the root-mean-square standard deviation (RMSSD) (Sharma, 1996), and the mean square distance (MSD) (Xie and Beni, 1991), which are commonly used in clustering studies to evaluate compactness of a cluster. The smaller the RMSSD or MSD, the better the compactness. It is illustrated in Figure 3 that the compactness of the "OTHER" class is significantly worse than target classes. Notably, the class with the second-worse compactness is the "MISC" class, i.e., the "miscellaneous" class, which is also a type of the Universum class.

Given that the Universum class has the highest number of samples in the datasets we examined, it should be represented best through training and hence formulate the most compact cluster in the representation space for the test set. Yet, our empirical findings paint a different picture. Then, an interesting question was raised: **Why does the Universum class, despite being the largest, exhibit the worst compactness in its learned representations?**

Our research indicates that the answer is rooted in its **inherent inter-class pattern**. The Universum class is defined as a cluster of samples that do not belong to any of the predefined target classes. Considering human annotation practices, an entity is labeled as *Location* when it aligns with established patterns of *Location* entities. In contrast, a sample is labeled as *Others* not due to intra-class patterns specific to the *Others* class, but because it fails to conform to the patterns of *Location*, *Person*, or *Organization*. Consequently, current classification models are designed to recognize intra-class patterns and unable to discern the inherent inter-class predictive rule of the Universum class, which will result in the poorer compactness of the Universum class.

Additionally, from the lens of representation learning, the Universum class essentially encapsulates "everything else" in a given task. Thus, no

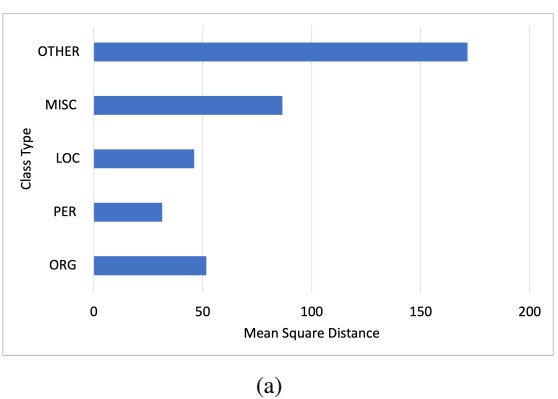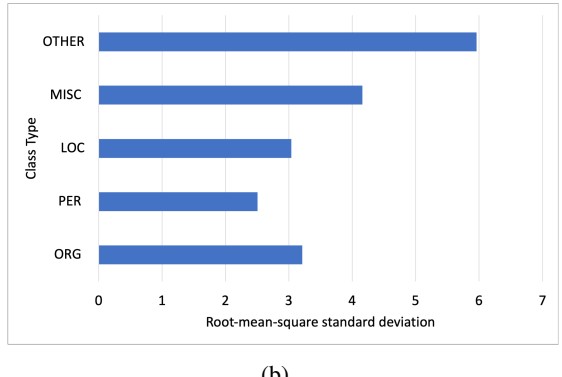

(a)                  (b)

Figure 3: The compactness evaluation of the Universum class and target classes of the test data of NER task.

matter the volume of this class, it's implausible to capture every nuance and pattern inherent to such a broadly defined class. This insight also sheds light on why, despite its significant size, the Universum class demonstrates the least compactness in its representations.

## B  Related Works

Continuing from previous discussions, we delve further into the related works on closed boundary learning methods here.

### B.1  Closed Boundary Learning Methods

Closed boundaries are often adopted in research fields of out-of-distribution (OOD) detection (Gomes et al., 2022; Ren et al., 2021; Chen et al., 2020), open set recognition (Zhang et al., 2021; Liu et al., 2020), anomaly detection (Zong et al., 2018), and outlier detection (Sharan et al., 2018; Sugiyama and Borgwardt, 2013). We borrow the term "generalized OOD detection" from (Yang et al., 2021b) to encapsulate these problems and discern their differences from our proposed classification with the Universum class problem.

### B.1.1  Difference in Problem Setting

Classification tasks can be categorized into problems based on closed-world assumption and open-world assumption (Yang et al., 2021b). The generalized OOD detection is treated under the open-world assumption, while the classification problem with the Universum class is treated under the closed-world assumption. In addition, the OOD samples are not available in the training data in generalized OOD detection, whereas a considerable number of Universum samples are included in the training data in our problem setting. The information of ex-

isting Universum samples is important to generate accurate decision boundaries in our problem.

### B.1.2  Difference in Methodology

By definition, the OOD detection problem assumes that the training data do not contain any OOD samples. However, a branch of the OOD studies, known as outlier exposure (Katz-Samuels et al., 2022; Ming et al., 2022; Yang et al., 2021a; Thulasidasan et al., 2021; Mohseni et al., 2020; Hendrycks et al., 2018), introduces auxiliary outlier data during training. The introduced auxiliary data makes it close to the format of our raised classification problems with the Universum class. However, outlier exposure methods are not suitable for our problem. The outlier exposure method mostly adopts a two-step approach that consists of multi-class classification and OOD identification. Such two-step approach will suffer from error propagation problem. In addition, the OOD identification step distinguishes OOD and ID samples based on a score obtained by cross entropy or energy function. However, both cross entropy and energy function are monotonically varying. As a result, the decision boundary derived from a threshold score of the monotonically varying function is an open boundary, which leaves the heterogeneity and representativeness issues we pointed out in this paper still unresolved.

From a methodological point of view, our work is also different from the works in generalized OOD using closed boundaries. In generalized OOD studies, the closed boundaries are formulated by the classic density-based method (Pidhorskyi et al., 2018; Hu et al., 2018), one-class classification method (Reiss et al., 2021; Ruff et al., 2018), or distance-based method (Gomes et al., 2022; Sun

et al., 2022; Zhang et al., 2021; Zaeemzadeh et al., 2021; Shu et al., 2020). The distance-based methods are limited to spherical boundary shapes but our method can generate arbitrary shape boundaries. The one-class classification method formulates only one closed boundary between positive and negative samples while our work generates closed boundaries for all target classes. Finally, only positive samples are used to learn decision boundaries in density-based method, while both target class samples and Universum samples are used in our work.

## C   Defining the Universum Class

Universum class exists in many tasks and datasets as we summarized in Table 1. Notably, the Universum class has various names such as *other* and *miscellaneous*, etc. In sentiment analysis, the *neutral* class can be considered as the Universum class because the word *neutral* is defined as "having no strongly marked or positive characteristics or features", which means the *neutral* class is a collection of all samples without strong emotions. Similarly, the *no relation* class in the relation extraction task can be considered as the Universum class.

## D   Implementation Details

### D.1   Baseline Models

We reproduce the baseline models based on the officially released source code, and apply closed boundary learning on the source code. All reported results are the average of three runs. It should be noted that some results of baseline models are slightly different from those given in the original papers due to the variations in random seeds and package versions when reproducing baseline models from their officially released codes. Nevertheless, baseline models and models with closed boundary learning are fairly compared in our work under the same random seed and deep learning environment.

In the six baseline models we selected, different results based on multiple language models are often reported in one work. We choose one of the pretrained models used in each work and reproduce the baseline models. The pretrained language model we used in each baseline and their reported results are summarized in Table 6.

### D.2   Training Process

During pretraining process, all parameters of the original model $\theta$ are learned. We employ GMM estimation on training data after pretraining and obtain the initial value of $\boldsymbol{\mu}_i$, $\boldsymbol{\Sigma}_i$, and $\pi_{ij}$, where $i \in \{1, 2, \cdots, n-1\}$, $j \in \{1, 2, \cdots, m\}$. $n$ is the number of classes, and $m$ is the number of GMM components. Through preliminary experiments, we observed that the number of GMM components has a minimal impact on our model's performance. Thus, we typically select $m = 4$ in our experiments. The threshold values $\xi_i$ is initialized around the $a$ quantile of $p(\mathbf{h}_k, C_i)$ values ($k \in \{0, 1, \cdots, N_i - 1\}$), where $a$ is the accuracy or F1 score of the original model. With our inter-class rule-based probability estimation for the Universum class, we obtain $[p(\mathbf{h}_k, C_1), \cdots, p(\mathbf{h}_k, C_{n-1}), p(\mathbf{h}_k, U)]$. Then, the original model parameters $\theta$, GMM parameters $\boldsymbol{\mu}_i$, $\boldsymbol{\Sigma}_i$, $\pi_{ij}$ and threshold values $\xi_i$ are learned by cross-entropy loss and our proposed boundary learning loss.

We use NVIDIA RTX A5000 GPUs to run the experiments and the model parameters are mostly follow the original baseline models.

### D.3   Robustness Evaluation

We evaluate the robustness of the model based on TextFlint (Wang et al., 2021), a robustness evaluation toolkit for NLP tasks. There are two kinds of transformations provided by TextFlint to generate the robust evaluation dataset, namely universal transformation and task-specific transformation. We adopt two universal transformations and two task-specific transformations to the test set of NER and RE task and generate four robustness evaluation datasets for each task. The terms of different transformations are explained below.

- "SpellingError": Universal transformation. Brings slight errors to words in the test samples.

- "AppendIrr": Universal transformation. Add irrelevant information to test samples.

- "CrossCategory": Task-specific transformation for NER. Replace the entity spans with substitutions from a different category.

- "OOV": Task-specific transformation for NER. Replace the entity spans with substitutions out of vocabulary.

| Method | Pretrained Model | Reported F1/accuracy |
|--------|-----------------|---------------------|
| SpanNER (Fu et al., 2021) | BERT-base | 92.28 |
| BS (Zhu and Li, 2022) | RoBERTa | 93.65 |
| A-GCN (Tian et al., 2021) | BERT-base | 89.16 |
| TaMM (Chen et al., 2021) | BERT-base | 89.18 |
| AC-MIMLLN (Li et al., 2020) | Glove | 76.42 |
| SCAPT (Li et al., 2021b) | BERT-base | 85.24 |

Table 6: The pretrained models chosen for each baseline model and the corresponding F1 score/accuracy reported in the original paper.

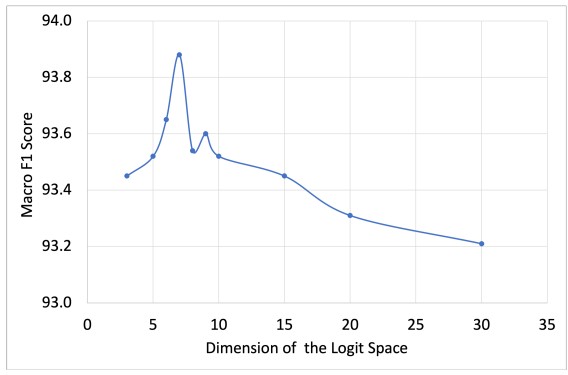

Figure 4: Impact of the last layer dimension on the accuracy of the model.

- "InsertClause": Task-specific transformation for RE. Change sample sentences by appending adjuncts from the aspect of dependency parsing.

- "SwapEnt": Task-specific transformation for RE. Swap the named entities in a sentence into entities of the same type.

Specifically, the models are trained and validated on the original training set and validation set, but the test set is transormed into the robustness evaluation dataset by the transformations proposed by TextFlint. Then, modelw are tested on the transformed test set.

## E The Impact of the Final Layer Dimension

The last layer's dimensionality can affect the performance of the model. Recalling the classic Hughes phenomenon (Hughes, 1968) that the model accuracy is monotonically increasing first and then monotonically decreasing with the dimension of data, the dimension of the final layer may be chosen to boost model performance.
We investigate the effect of last layer dimension on the accuracy of the model on the SpanNER (Fu et al., 2021) with closed boundary learning and present the result in Figure 4. The F1 score of the test set grows with increasing of dimensions and reaches a maximum value of 93.88 when the dimension is seven, and then decreases with the dimension. The trend fits well with the Hughes phenomenon (Hughes, 1968). Our method is quite robust with the dimension and the overall result of SpanNER + COOLU reported in Table 2 is set as ten rather than the optimal value.