# OpenReview forum: "Closed Boundary Learning for Classification Tasks with the Universum Class"
_EMNLP/2023/Conference — EMNLP 2023 Findings_

### Official Review · Reviewer_RZMD · 2023-08-03

**Soundness:** 3

**Excitement:**

4: Strong: This paper deepens the understanding of some phenomenon or lowers the barriers to an existing research direction.

**Paper Topic And Main Contributions:**

This paper proposes a closed-boundary classification method to deal with the miscellaneous class with many clusters.

**Reasons To Accept:**

The proposed approach of building closed boundary classifier to deal with Universum class is interesting.

The proposed method is novel.

**Reasons To Reject:**

I am surprised that a closed boundary method can do well. Is it because the NER and RE tasks are particularly amenable to such a method? For supervised classification, I am not sure. Can you include some traditional supervised classification methods.

OOD detection researchers have tried different types of approaches to build closed boundary classifiers. In NLP, there are many papers, e.g., DOC: Deep Open Classification of Text Documents. EMNLP-2017; Deep open intent classification with adaptive decision boundary, AAAI-2021; KNN-contrastive learning for out-of-domain intent classification. ACL2022; Adversarial Self-Supervised Learning for Out-of-Domain Detection, NAACL-2021.

Your baselines should include some SOTA OOD detection methods. I think kNN with a pre-trained feature extractor should do quite well too.

**Reproducibility:**

3: Could reproduce the results with some difficulty. The settings of parameters are underspecified or subjectively determined; the training/evaluation data are not widely available.

**Reviewer Confidence:**

4: Quite sure. I tried to check the important points carefully. It's unlikely, though conceivable, that I missed something that should affect my ratings.

---

> ### Author Rebuttal · Authors · 2023-08-29
>
> Dear Reviewer RZMD,
>
> We sincerely appreciate your thorough review, constructive feedback, and valuable support on our paper. Your insights have been instrumental in helping us improve the quality of our work. Please find below a summary of our responses to your comments and concerns.
>
> ---
> **[C1]: I am surprised that a closed boundary method can do well. Is it because the NER and RE tasks are particularly amenable to such a method? For supervised classification, I am not sure. Can you include some traditional supervised classification methods.**
>
> **[A1]**: Thank you for raising this question. To address your concern, we provide a detailed explanation below.
>
> The six classification methods we evaluated, spanning the RE, NER, and sentiment analysis tasks, are all supervised classification methods. Specifically:
> * The NER task classifies entity spans into different entity classes.
> * The RE task classifies the entity pairs into various relation classes.
> * The sentiment analysis task classifies sentences into different sentiment classes.
>
> **While the strategy for representation learning differs among these methods, they adhere to the standard procedure typical of supervised classification methods**: they learn representation, then map representation space into logits, subsequently pass logits through a softmax function to derive probabilities for each class, and finally employ the cross-entropy loss for optimization.
>
> To answer your question about **why a closed boundary method can do well**, we believe the answer lies in two aspects:
> * The Universum class exhibits very different properties from the target classes. Target class samples follow intra-class patterns: considering the human annotation process, an entity is labeled as "*Location*" because it aligns with the common patterns of "*Location*" entities. **In contrast, the Universum samples follow a distinct inter-class pattern: an entity is labeled as "*Others*" not because it aligns with common patterns of "*Others*" entities, but because this entity doesn't align with patterns of "*Location*", "*Person*", and "*Organization*" classes**. Overlooking the special properties of the Universum class can lead to problems such as overfitting, diminished robustness, and misclassification of unseen samples, as we illustrated between lines 86 to 121 in our paper.
> * On the other hand,  the question arises: Why do closed boundary methods in OOD detection fail to improve the performance of these classification tasks? The primary reason is that they are originally designed for a different problem setting. This mismatch leads to problems such as error propagation inherent to two-step methods, instability from manually set thresholds, and the underutilization of Universum class samples, all leading to poor performance. In contrast, our framework introduces the "Inter-Class Rule-Based Probability Estimation for the Universum Class", which allows for the joint learning of both the Universum class and target classes. Additionally, we propose the boundary learning loss to facilitate the learning of closed boundaries. We also transform the manually set threshold value into a learnable parameter of the neural network.
>
> ---
> **[C2]: Include some OOD detection methods as baselines.**
>
> **[A2]**: We highly value your constructive suggestion. In light of this, we have incorporated **additional baselines** for evaluation. Specifically, we assessed both the ADB [1] and KNN-contrastive learning [2] methods. Experimental findings consistently demonstrate that our method achieves superior performance compared to these baselines. We are willing to include more baseline experiments in the revised manuscript.
>
> | | SpanNER | A-GCN | AC-MIMLLN |
> |---------|----------|----------|----------|
> | ADB | 87.81 | 85.99 | 71.78 |
> | KNN-Contrastive | 89.08 | 86.32 | 72.16 |
> | COOLU |   **93.50** | **89.33** | **77.35** |
>
> ***Table 1**: Performance comparison with additional baselines.*
>
> [1] Zhang, Hanlei, Hua Xu, and Ting-En Lin. "Deep open intent classification with adaptive decision boundary." Proceedings of the AAAI Conference on Artificial Intelligence. Vol. 35. No. 16. 2021.
>
> [2] Zhou, Yunhua, Peiju Liu, and Xipeng Qiu. "KNN-contrastive learning for out-of-domain intent classification." Proceedings of the 60th Annual Meeting of the Association for Computational Linguistics (Volume 1: Long Papers). 2022.
>
> ---
> We deeply appreciate your insights, which have significantly enriched the quality of our paper. We look forward to hearing your feedback!

---

### Official Review · Reviewer_gG1E · 2023-08-04

**Typos Grammar Style And Presentation Improvements:** 1) The authors could abbreviate the c…
**Soundness:** 3

**Excitement:**

3: Ambivalent: It has merits (e.g., it reports state-of-the-art results, the idea is nice), but there are key weaknesses (e.g., it describes incremental work), and it can significantly benefit from another round of revision. However, I won't object to accepting it if my co-reviewers champion it.

**Paper Topic And Main Contributions:**

This paper formulates a new question of distinguishing the Universum class and the interest classes in classification tasks. The authors aim to leverage closed decision boundaries with a rule-based probability estimation strategy to classify the universum and target class. The experiments on the proposed COOLU demonstrate the comprehensive improvement over the BS, A-GCN, TaMM, AC-MIMLLN, SpanNER, and SCAPT benchmarks.

**Questions For The Authors:**

1) How does the COOLU improving NER and ACSA tasks compare to the method improving RE tasks? More improvement can find the NER and ACSA tasks.
2) I will suggest a more detailed experimental setting such as a case study and visualization. It may be better to analyze how to discriminate the multi-hop relations (universum or targe class) and their heterogeneity and diversity in RE.



**Reasons To Accept:**

1)  The paper is clear and well-written. I appreciate the comprehensive elaboration with four research questions and arguments about the beneficial classification performance and robustness of COOLU for universum class and other classes.
2)  This paper shows the previously ignored connections between the universum class with inherent properties and the misclassification of samples of interest in diverse classification tasks.
3)  The idea of using closed decision boundaries for classification tasks is well-motivated, and this paper is easy to follow.

**Reasons To Reject:**

1) I understand the insight of GMM with inter-class rule-based probability estimation for unversum classification. But it seems GMM is not an ideal module, which is constraint on the definite and low dimensionality of the class setting.
2) A lack clarity about the mapping between the threshold value, the scale of data, task types, number of classes and the performance.


**Reproducibility:**

4: Could mostly reproduce the results, but there may be some variation because of sample variance or minor variations in their interpretation of the protocol or method.

**Reviewer Confidence:**

3: Pretty sure, but there's a chance I missed something. Although I have a good feel for this area in general, I did not carefully check the paper's details, e.g., the math, experimental design, or novelty.

---

> ### Author Rebuttal · Authors · 2023-08-29
>
> Dear Reviewer gG1E,
>
> We sincerely appreciate your thorough review and valuable feedback on our paper. Your insights have been instrumental in helping us improve the quality of our work. Please find below a summary of our responses to your comments and concerns.
>
> ---
> **[C1]: GMM is not an ideal module, which is constraint on the definite and low dimensionality of the class setting.**
>
> **[A1]**: Thank you for drawing attention to the potential constraints of GMM, especially in the context of definite and low-dimensionality class settings.
>
> Our framework is applied after the final layer of classification models, which is a linear layer that maps data from a high-dimensional space down to a dimensionality equal to the number of classes of the task. Consequently, the input dimension for GMM is essentially fixed by the number of classes. Additionally, given that most classification tasks involve a relatively small number of classes, GMM's limitations regarding low dimensionality become less critical for our framework. Finally, in our framework, the neural network dynamically updates both the GMM parameters and the data distribution of the GMM input in the feature space throughout the training process, which may also enhance the performance of GMM.
>
> However, we concur that as the number of classes grows significantly, GMM's inherent constraints could challenge our framework's effectiveness. Your feedback highlights the importance of exploring better alternatives in future works. We appreciate this perspective.
>
> ---
> **[C2]: A lack clarity about the mapping between the threshold value, the scale of data, task types, number of classes and the performance.**
>
> **[A2]**: Thank you for highlighting this important point. We understand that these factors' effect on performance needs a clearer exposition in our work. Here's a deeper analysis:
>
> * **Threshold Value**: Contrary to a manually selected parameter, the threshold value is a learnable parameter in the model. This treatment prevents us from distinctly evaluating its impact on performance.
> * **Scale of Data**: Our current experiments utilize the full data supervised training setting, aligning with the classification tasks we evaluated. However, we would like to incorporate experiments with varied data proportions (e.g., 25%, 50%, 75%) in our revised version, anticipating insightful findings.
> * **Task Types**: As shown in Table 1 of the paper, the proportion of the Universum class fluctuates across tasks. Notably, the RE task has the lowest proportion at 17.4%, compared to 90% and 43.4% in other tasks. This disparity might primarily account for the least improvement we observed on the RE task.
> * **Number of Classes**: As the number of classes increases, GMM's inherent dimensionality constraints become more pronounced, potentially impacting performance. This may be validated in the RE task, which, with the largest number of classes, shows less improvement with our method.
>
> We notice that while most of the above information is present in the paper, it hasn't been systematically collated and analyzed. **Your feedback is insightful, and in response, we will present a clearer mapping between these factors, aiding readers in understanding their influence on the model's performance better**.
>
> ---
> **[Q1]: How does the COOLU improving NER and ACSA tasks compare to the method improving RE tasks? More improvement can find the NER and ACSA tasks.**
>
> **[A1]**: Thank you for the question. We have provided an analysis of this issue in "Task Types" and "Number of Classes" in answers of **C2**.
>
> ---
> **[Q2]: I will suggest a more detailed experimental setting such as a case study and visualization.**
>
> **[A2]**: Thank you for your constructive feedback. We will integrate the visualisation and case study in the revised manuscript.
>
> ---
> **[Response to Typos and Presentation Improvement]**
>
> Thank you for your thorough examination of our work and for highlighting the typos and suggestions on the presentation. We will enhance the clarity and presentation of our paper following your suggestions.
>
> ---
> We deeply appreciate your insights, which have significantly enriched the quality of our paper. We look forward to hearing your feedback!

---

### Official Review · Reviewer_G4Yh · 2023-08-05

**Soundness:** 3

**Excitement:**

3: Ambivalent: It has merits (e.g., it reports state-of-the-art results, the idea is nice), but there are key weaknesses (e.g., it describes incremental work), and it can significantly benefit from another round of revision. However, I won't object to accepting it if my co-reviewers champion it.

**Paper Topic And Main Contributions:**

The paper points out a previously underexplored problem of the Universum class ("other", "miscellaneous") in multi-class classification problems. New approach based on N-pair loss pretraining, GMM and new loss function is proposed, which produced closed decision boundaries around remaining (not Universum) classes. The Universum class is specially handled (lack of closed decision boundary) and treated as a remaining ("out-of-distribution") class. The experiments on NER, relation extraction and sentiment analysis demonstrate that the framework offers approx. 1% improvement in F1/acc measures.

**Questions For The Authors:**

A. How the gradients are propagated through loss function, GMM etc. to train the model?
B. Why do you think that comparison only with the baseline (you treat OECC as an ablation) is sufficient to show the usefulness of your approach?
C. To what extent training the model with your method is more computational expensive than the training of the baseline?
D. What statistical test were used?

**Reasons To Accept:**

- interesting idea and relatively new problem
- formulation of a new loss function producing closed decision boundaries (unfortunately, some parts of it are not clear -see Reasons to Reject)

**Reasons To Reject:**

- Overall, it is difficult to understand how actually the whole framework works (lack of a pseudocode or a schema). After pretraining, GMMs is estimated (with Expectation-Maximization? = it is not specified). Then the probabilities from GMM are used in a loss function to learn thresholds ξ. Later, somehow cross-entropy loss is applied on class probabilities which are computed basing on GMM with hard thresholds (Eq. 4) to update the models' parameters. How the gradients are propagated through loss function with probabilities computed using hard thresholds (for Universum class) and GMM (trained by EM?) to finally get to the classification model?
- Experiments.
	- The authors compare their approach by extending with it several models. However, apart from the baseline only one other method for outlier exposure (which is unsuitable for universum classes according to the authors) is taken into account. Other related approaches are not studied.
	- The authors performed some statistical tests, but they do not specify which one. Were the test assumptions properly checked?
	- The authors formulate a research question "does COOLU provide a more reasonable way of learning representations and classifiers". What "reasonable way" of learning classifiers actually means? This statement is very imprecise.
	- Lack of some ablation studies, e.g. how well the method works without N-pair loss pretraining?
	- Some experimental details are not clear. E.g. How the number of GMM components was selected? What is the time complexity of this approach? (how the training time of the new method compare with the training time of the baseline?)

- Motivation
	The authors claim the lack of representativeness of Universum class in training data. However, in most datasets used in the experimental study such class is quite large, hence having abundant representation in the data. That would mean that basically all the classes are not represented well enough in the training data. The authors probably have in mind the fact that Universum class is an aggregation of "all possible implicit classes" but still this claim about lack of representativeness is only supported by a visualization (which is not based on any theoretical model or real data).

**Reproducibility:**

3: Could reproduce the results with some difficulty. The settings of parameters are underspecified or subjectively determined; the training/evaluation data are not widely available.

**Reviewer Confidence:**

3: Pretty sure, but there's a chance I missed something. Although I have a good feel for this area in general, I did not carefully check the paper's details, e.g., the math, experimental design, or novelty.

**Typos Grammar Style And Presentation Improvements:**

- "Figure 2 is an illustration of formulating an arbitrary shape boundary in a three- dimensional space." = it is a 2d shape. The third dimension is the value of prob. density.
- grammar "An intuitive way to incorporate the above rules is a two-step method consists of Universum class detection and target classes classification
- (6) presents two separate symbols,  remove the bracket


In 3.3.1 authors introduce 2 rules to estimate probabilities. Later, they claim that using this rules (which work in a mutually exclusive conditions) would actually result in a pipeline method which has the issue of error propagation. In fact, both rules can be easily rephrased into only one rule and their direct application does not have to lead to error propagation. But I agree with the motivation about facilitating the learning of the
neural network.

---

> ### Author Rebuttal · Authors · 2023-08-29
>
> Dear Reviewer G4Yh,
>
> We sincerely appreciate your thorough review and valuable feedback on our paper. Your insights have been instrumental in helping us improve the quality of our work. Please find below a summary of our responses to your comments and concerns.
>
> ---
> **[C1]: it is difficult to understand how actually the whole framework works**
>
> **[A1]: We genuinely appreciate your feedback, emphasizing the need for a clearer and more integrated presentation of our method. We recognize this gap and are committed to enhancing the clarity in the revised manuscript.**
>
> To address this, we provide a succinct breakdown of our framework's workings:
> * **Initialization Post-Pretraining**: After pre-training, we employ the GMM for each target class. The Expectation-Maximization (EM) algorithm is employed to set the initial values for the GMM parameters $\mu_i$, $\Sigma_i$, and $\pi_{ij}$. As we transition to the training phase, the parameters of GMM are treated as learnable parameters. Contrary to traditional methods using the EM algorithm for continuous updates, these parameters are dynamically updated by the neural network throughout the training process.
> * **Probability Estimation**: Probability distributions of target classes are estimated using GMM, as articulated in Eq. 2. The probability distribution of the Universum class is computed through our Inter-Class Rule-Based Probability Estimation method, which is represented in Eq. 4.
> * **Training Optimization**: During the training process, we use a combined loss function, summing the cross-entropy loss with the boundary learning loss for optimization.
>
> Given your feedback, we realize that while each component of our framework was detailed in the paper, the absence of an overall description might have fragmented the reader's understanding. To rectify this, we commit to enrich the revised version with a schematic representation of our framework.
>
> **In response to your question on gradient propagation**: updating mechanics for the GMM parameters are introduced in the "Initialization Post-Pretraining" above, where we emphasize that these parameters are dynamically updated by the neural network rather than the EM algorithm. Addressing the non-differentiability points in Eq. 4, similar to the ReLU function at 0, in practical applications, exact non-differentiable points, like 0 for ReLU or the boundary in Eq. 5, are not likely reached due to precision constraints. Moreover, near this boundary, the gradients of the components in Equation 4 exhibit close resemblances, which curbs the potential for the model to be steered in suboptimal directions near the boundary.
>
> ---
> **[C2]: Questions and concerns with experiments.**
>
> **[A2]: We greatly value your insightful feedback on our experiments. Please find our detailed responses to each of your points below:**
>
> * Comparison with Other Methods:
>
> We appreciate your suggestion to broaden our experimental comparisons. In light of this, we have incorporated **additional baselines** for evaluation. Specifically, we assessed both the ADB [1] and KNN-contrastive learning [2] methods. Experimental findings consistently demonstrate that our method achieves superior performance compared to these baselines.
>
> | | SpanNER | A-GCN | AC-MIMLLN |
> |---------|----------|----------|----------|
> | ADB | 87.81 | 85.99 | 71.78 |
> | KNN-Contrastive | 89.08 | 86.32 | 72.16 |
> | COOLU |   **93.50** | **89.33** | **77.35** |
>
> ***Table 1**: Performance comparison with additional baselines.*
>
> [1] Zhang, Hanlei, Hua Xu, and Ting-En Lin. "Deep open intent classification with adaptive decision boundary." Proceedings of the AAAI Conference on Artificial Intelligence. Vol. 35. No. 16. 2021.
>
> [2] Zhou, Yunhua, Peiju Liu, and Xipeng Qiu. "KNN-contrastive learning for out-of-domain intent classification." Proceedings of the 60th Annual Meeting of the Association for Computational Linguistics (Volume 1: Long Papers). 2022.
>
> * Statistical Test:
>
> We used the paired t-test for our statistical evaluation and ensured its assumptions were met.
>
> * Definition of "reasonable way":
>
> Thank you for pointing out this ambiguous phrasing. By "reasonable way," we aimed to convey that COOLU offers a method that is capable to improve both the accuracy and robustness of classification models. We will refine this statement in the revised version to better communicate our intention.
>
> * Ablation study on N-pair loss:
>
> The ablation study of N-pair loss is provided in Appendix E. Our method relies on the pre-trained feature space for GMM initialization, thus, removing the pretraining step isn't feasible. However, to transparently demonstrate the effectiveness of our approach, we integrate N-pair loss pretraining into the baseline method. This is done to clarify that the observed performance improvements stem from the unique attributes of our proposed method, rather than merely the addition of the pretraining step.
>
> * Experimental details:
>
> We appreciate the emphasis on clarity of experimental details. The choice of GMM components was informed by our preliminary experiments. We observed that the number of GMM components had a minimal impact on our model's results. Moreover, there wasn't a need for an excessive number of components. Thus, we set four as the choice for our method. In terms of training time, our method took additional pre-training step (~20% of the total training epochs) and one more epoch specifically for the GMM parameters' initialization, in contrast to the baseline. We will ensure these details are clearly presented in the revised version.
>
> ---
> **[C3]: Concern on motivation**
>
> **[A3]: We apologize for any confusion stemming from our paper's presentation and value your insight on this matter. The crux in addressing your concern lies in recognizing the inherent inter-class pattern of the Universum class. To elucidate this further, we would like to address your concern through experimental results and theoretical explanations.**
>
> From an experimental standpoint, we understand your view that a larger sample size would yield better representativeness of the class.   Given that the Universum class is the most populous in the datasets we examined, it should be represented best through training and hence formulate the most compact cluster in the representation space for the test set. **Yet, our empirical findings paint a different picture.** As depicted in Figure 1 (b) of the paper, the averaged compactness of the Universum class is significantly worse, being over two times less compact than most target classes. The experimental details can be found in Appendix A.
>
> Then, an interesting question was raised "**Why does the Universum class, despite being the largest class, exhibit the worst compactness in its learned representations?**" Our research indicates that the answer is rooted in its **inherent inter-class pattern**. The Universum class is defined as a cluster of samples that do not belong to any of the predefined target classes. Considering how humans recognize this class, during dataset annotation, an entity is labeled as "*Location*" because it aligns with the patterns of "*Location*" entities. In contrast, another sample is labeled as "*Others*", not because it follows the inner patterns of the "*Others*" entities, but because it doesn't match patterns of "*Location*", "*Person*", and "*Organization*". Consequently, neural networks are better suited to recognize this class by its lack of alignment with target classes (its inter-class pattern) rather than by seeking specific intra-class patterns from its samples.
>
> From the lens of representation learning, the Universum class essentially encapsulates "**everything else**" in a given task. Thus, no matter the volume of this class, it's implausible to capture every nuance and pattern inherent to such a broadly defined class. This insight sheds light on why, despite its significant size, the Universum class demonstrates the least compactness in its representations.
>
> Furthermore, we delve into the consequences of not giving the Universum class specialized treatment, such as overfitting and diminished robustness between lines 86 to 107 in our paper. Thank you for your feedback, We will refine our motivation for greater clarity in the revised revision
>
> ---
> **[Respond to Questions and typos]**
>
> Thank you for your informative questions. These questions are already addressed in our previous responses.
>
> Thank you for your thorough examination of our work and for highlighting the typos. We commit to conducting a meticulous review of the manuscript and rectifying all typos in our revised version.
>
> ---
> We deeply appreciate your insights, which have significantly enriched the quality of our paper. We look forward to hearing your feedback!

---

### Meta-Review · Area_Chair_M4qd · 2023-09-15

**Recommendation:** 4

**Metareview:**

This paper addresses the issue of distinguishing the "Universum class" from interest classes in classification tasks. It introduces a closed-boundary classification method with a rule-based probability estimation approach to classify the Universum and target classes. The proposed COOLU method outperforms several benchmarks like BS, A-GCN, TaMM, AC-MIMLLN, SpanNER, and SCAPT. The approach handles the miscellaneous class effectively, offering approximately a 1% improvement in F1/accuracy measures in NER, relation extraction, and sentiment analysis experiments.

The problem addressed is important in many real-world scenarios, and the results presented in the paper are promising. However, the main issue with the work lies in its clarity, as indicated in the discussion, which required several additional details to fully appreciate the paper and this research.

---

### Decision · Program_Chairs · 2023-10-07

**Decision:**

Accept-Findings

**Comment:**

This paper addresses the issue of distinguishing the "Universum class" from interest classes in classification tasks. It introduces a closed-boundary classification method with a rule-based probability estimation approach to classify the Universum and target classes. The proposed COOLU method outperforms several benchmarks like BS, A-GCN, TaMM, AC-MIMLLN, SpanNER, and SCAPT. The approach handles the miscellaneous class effectively, offering approximately a 1% improvement in F1/accuracy measures in NER, relation extraction, and sentiment analysis experiments.

The problem addressed is important in many real-world scenarios, and the results presented in the paper are promising. However, the main issue with the work lies in its clarity, as indicated in the discussion, which required several additional details to fully appreciate the paper and this research.